# Is Oral Iron and Folate Supplementation during Pregnancy Protective against Low Birth Weight and Preterm Birth in Africa? A Systematic Review and Meta-Analysis

**DOI:** 10.3390/nu16162801

**Published:** 2024-08-22

**Authors:** Yibeltal Bekele, Claire Gallagher, Mehak Batra, Don Vicendese, Melissa Buultjens, Bircan Erbas

**Affiliations:** 1School of Psychology and Public Health, La Trobe University, Melbourne, VIC 3086, Australia; y.bekele@latrobe.edu.au (Y.B.); m.batra@latrobe.edu.au (M.B.); m.buultjens@latrobe.edu.au (M.B.); 2School of Public Health, Bahir Dar University, Bahir Dar 79, Ethiopia; 3School of Population and Global Health, The University of Melbourne, Melbourne, VIC 3010, Australia; claire-gallagher@live.com (C.G.); don.vicendese@unimelb.edu.au (D.V.); 4School of Computing, Engineering and Mathematical Sciences, La Trobe University, Melbourne, VIC 3086, Australia

**Keywords:** low birth weight, preterm birth, iron supplementation, folate supplementation, iron–folic acid supplementation, Africa

## Abstract

Background: Despite recent evidence demonstrating iron and folate supplementation reduces the risk of low birth weight and preterm births, synthesis of the evidence is not sufficient to understand their impacts in Africa. Method: MEDLINE, PsycINFO, Embase, Scopus, CHINAL, Web of Science, Cochrane databases, and Google Scholar were searched for the published and grey literature. Either iron-only, folate-only, or iron–folic acid (IFA) oral supplementation during pregnancy was the primary exposure/intervention. The focus of this review was low birth weight and preterm births in the African region. Qualitative synthesis, meta-analysis, and subgroup analysis were employed. Results: In the qualitative synthesis (*n* = 4), IFA supplementation showed a positive impact on reducing preterm birth. Additionally, the meta-analysis showed that IFA and iron-only supplementation reduced the odds of low birth weight by 63% (OR 0.37; 95% CI: 0.29, 0.48) and 68% (OR 0.32; 95% CI: 0.21 to 0.50), respectively. Conclusion: Both iron-only and IFA supplementation are effective in reducing the risk of low birth weight in Africa. There is also promising evidence suggesting a potential reduction in preterm births. Consequently, further research is needed, particularly targeting high-risk groups such as women residing in rural areas with limited support and low levels of literacy.

## 1. Introduction

Prematurity and low birth weight persist as the primary causes of death among newborns and children under five years of age, especially in regions like Africa [1,2]. Even now, more than one-third of global low-birth-weight cases and preterm births occur in Africa [3,4]. Despite a reduction in the rate of low birth weight and preterm birth over the past two decades, the rate of reduction is much slower in low- and middle-income countries, particularly in Africa [5,6]. Studies in Africa reveal ongoing challenges. For example, in Ethiopia, low birth weight prevalence was reported at 21% [7]. In Ghana, the prevalence of preterm birth was 37.3% [8]. These rates are distant from the countries’ promise to reduce the rate below 12 per 1000 live births by the end of 2030 [9]. 

Preterm birth and having a low birth weight exert significant adverse short- and long-term health, economic, and social impacts [10,11]. These conditions pose risks such as poor academic performance [12], chronic medical problems [13], reduced productivity [14], and increased mortality (especially during neonatal and infant periods) [15,16]. Beyond these consequences, preterm and low birth weights impose substantial economic losses in the African region. For instance, a study conducted in Sub-Saharan Africa reported that caring for low birth weight costs the patient up to USD 489 and the health care system USD 514 [17]. Similarly, a study conducted in Nigeria reported that the overall cost of hospital care for very low-birth-weight infants ranges from USD 211.1 to USD 1573.9 [18]. Another study in Rwanda reported that the direct medical cost of each preterm birth was USD 237.7, in total constituting around 28% of the country’s gross domestic product [19]. Hence, focusing on the prevention of low birth weight and preterm births could have a substantial impact on the public health burden in this region. 

Iron and folic acid supplementation during pregnancy plays a crucial role in supporting healthy growth and development [20] while also protecting against low birth weight and preterm birth [21]. The increased vascular volume and rapid growth and development of the foetus during pregnancy can lead to iron depletion [22,23]. Studies have shown that iron and folic acid deficiency during pregnancy increase the risk of low birth weight and preterm births [24,25]. To mitigate these impacts, the World Health Organisation (WHO) recommends a daily supplementation of 30 to 60 mg of iron and 0.4 mg of folate for all pregnant women throughout their pregnancy [26].

Despite worldwide efforts to increase supplementations of iron and folate [27,28], the uptake remains low. For example, a systematic review conducted in Sub-Saharan Africa in 2021 showed that compliance with iron folate supplementation was 39.2%, varying from 10.6% in Kenya to 79% in Mozambique [29]. Studies have reported that poor access to, and poor quality of antenatal care (ANC) services, healthcare financing, and lack of supplies are the main barriers to poor coverage and adherence in the region [30,31]. Similarly, a study conducted in Africa reported that the erratic supply in primary healthcare facilities and unaffordable costs of private pharmacies are the main barriers to service utilisation [32]. 

Currently, most African countries use both iron and folate supplementation as a main nutritional intervention to improve the health of mothers and their offspring [33,34,35]. The region faces significant challenges, including anaemia, severe nutritional problems, and poor maternal healthcare service utilisation [36,37,38,39]. Although iron and folate supplementation is widely implemented, there is limited evidence of its effectiveness in reducing adverse birth outcomes, specifically within the African context. 

Prior systematic reviews of randomised control trials and prospective cohorts have been conducted in 2012 and 2013 [16,40,41], which include a few African countries to assess the effects of iron-only and IFA supplementation on adverse birth outcomes and neonatal mortality; however, the findings were inconsistent across each review and there were differences in the inclusion and exclusion criteria. Reviews from 2012 [40,41] found that iron supplementation reduced the risk of low birth weight but had no effect on preterm birth. However, IFA supplementation was not associated with a reduction in either low birth weight or preterm births [41]. A systematic review conducted in 2013 [16] included oral iron-only supplementation and fortification as exposures. The review’s findings showed that iron-only supplementation reduced low birth weight with no measured effect on preterm birth. A more recent systematic review conducted in 2021 in low- and middle-income countries found that iron supplementation, compared to the placebo, and IFA supplementation, compared to folic acid alone, reduced the risk of low birth weight. Additionally, this review reported that multiple micronutrients (MMN) supplementation was more effective than iron, with or without folate, in reducing the risk of low birth weight [42]. Past reviews showed mixed findings and did not comprehensively include countries in Africa. For instance, a systematic review conducted in 2012 [40] included only one study from Gambia, and a 2013 review [16] similarly included just one study from Zimbabwe. This implies there is a need to further explore the effects of iron-only, folate-only, and IFA supplementation on preterm birth and low birth weight, particularly in Africa. 

Additionally, this systematic review focuses on selecting populations that have experienced a high prevalence of anaemia, poor quality and accessibility of health services, and low socioeconomic status. Unlike previous reviews, this systematic review will include both observational and experimental study designs conducted in the African region. 

To the best of our knowledge, this review is the first comprehensive systematic review and meta-analysis conducted in the African region. This review aims to assess the effects of iron-only, folate-only, and IFA supplementation on preterm birth and low birth weight. The findings of this review will help policymakers, program developers, and researchers working in the region. In addition, the findings can serve as valuable guidance for those formulating effective strategies and interventions in this context. The specific objectives are as follows:To synthesise available evidence on the relationship between oral iron and/or folate supplementation on low birth weight and preterm birth;To evaluate the dose–effect relationship and effect of duration of oral iron and/or folate supplementation on low birth weight and preterm birth;If the data allow, to conduct a subgroup analysis based on common sociodemographic characteristics

## 2. Materials and Methods

This systematic review and meta-analysis adhere to the Preferred Reporting Items for Systematic Review and Meta-Analysis (PRISMA-P) 2020 guidelines [43] (Appendix A). This review was registered in the PROSPERO International Prospective Register of Systematic Reviews (registration number: CRD42023452588). 

### 2.1. Selection Criteria

#### 2.1.1. Types of Studies

All randomized, non-randomized, and observational study designs conducted in Africa were eligible for inclusion. Abstracts without full articles, conference papers, and letters were excluded from the review.

#### 2.1.2. Population of Interest

Studies of healthy pregnant women in Africa were included in the review, regardless of age, parity, and gestational age. The participants were women aged 15 to 49 years, with approximately 67% of them residing in rural areas. 

#### 2.1.3. Exposure and Comparators

The exposure of interest was oral iron supplementation, oral folate supplementation, or oral IFA supplementation during pregnancy. We considered studies of any duration, timing, and dosage, provided that the iron and/or folate supplementation was compared with either a placebo, no iron, and/or folate supplementation. Studies that considered supplementation through fortification, in combination with other micronutrients, or measured serum iron or folate levels as the intervention were excluded from the review. 

### 2.2. Outcomes 

#### Birth Outcomes

Low birth weight (birth weight < 2500 g) [44];

Preterm birth (birth < 37 weeks) [44]. 

### 2.3. Context and Language

This review encompassed published articles in the English language conducted in Africa without specifying a particular time frame. 

### 2.4. Information Sources and Search Strategy

MEDLINE, PsycINFO, Embase, Scopus, CHINAL, Web of Science, and Cochrane electronic databases were searched to identify the relevant literature. Google Scholar and Google Advanced Search were also used to identify the grey literature. The database search was conducted on 29 August 2023 and the final Google Scholar search was conducted on 2 February 2024.

The search strategy was developed by the authors, with the support of La Trobe University’s research librarian, and is available in the online supplement (Appendix A). The database search was restricted to English-language publications.

### 2.5. Study Selection

Search results were exported to Covidence, and duplicates were removed. Title, abstract, and full-text screenings were conducted independently by two authors (Y.A.B. and C.G.) based on pre-established criteria. Any disagreements between authors were resolved by discussion or consultation with the senior author (B.E.).

### 2.6. Data Extraction 

The following information, such as the first name of the author, year of publication, study setting, design, population, rural population, sample size, types of exposure, types of outcomes, confounder adjustment, effect estimate, and 95% confidence intervals were extracted from each included study. Data extraction was independently conducted by two authors, Y.A.B. and C.G.

### 2.7. Quality Appraisal

Cochrane Effective Practice and Organisational Care (EPOC) [45] and the Newcastle–Ottawa Scale (NOS) risk of bias assessment tools were used to assess the quality of the studies [46]. Randomized and quasi-experimental designs were evaluated by EPOC, and studies were ranked as low (scores of 3 or less), medium (scores of 4 up to 6), and high risk (scores of 7 up to 10). The NOS risk of bias assessment tool was used for observational study designs and each article was classified into low (scores of 3 or less), medium (scores of 4 up to 6), and high risk of bias (scores of 7 up to 10). The risk of bias assessment was conducted independently by Y.A.B. and C.G., and disparities were resolved through discussion or with the senior author (B.E.). 

### 2.8. Data Synthesis and Meta-Analysis

Studies were grouped based on the exposure (iron only, folate only, or IFA supplementation) and the reported outcomes (preterm birth and low birth weight). Studies that reported multiple outcomes were included in both groups. 

Meta-analysis was conducted where possible using STATA 18 [47]. A forest plot was used to summarize the overall effect. We employed random-effects models in all meta-analyses. Heterogeneity across the studies was assessed using the *I*^2^ test, with a value greater than 50% considered as substantial heterogeneity, prompting further subgroup analysis. Subgroup analysis was conducted based on the country of the study and the study design. Publication bias was assessed using a funnel plot and Egger’s test. In Egger’s test, a *p*-value greater than 0.05 was considered sufficient statistical evidence against there being no publication bias. Odds ratios with 95% CIs were used to measure the effect sizes. We also conducted a narrative synthesis guided by the Synthesis Without Meta-analysis (SWiM) reporting guidelines [48] for studies that were not included in the meta-analysis.

## 3. Results 

### 3.1. Search Results and Characteristics of Included Studies

A total of 2174 non-duplicate records were retrieved, of which, 2095 studies were excluded during the title and abstract screening (Figure 1). During the full-text review, a further 57 studies were excluded, leaving 22 articles eligible for inclusion in the review. The primary reasons for excluding studies included a lack of interventions of interest, the absence of a relevant comparator, non-conforming study design, abstracts without full-text studies available, and outcomes unrelated or not relevant to this review’s focus. For instance, a study conducted by Toe et al. assessed lipid-based nutrient supplementation for interventions [49]. Studies undertaken by Caniglia et al. and Roberfroid et al. both assessed IFA supplementation as the comparison group [50,51]. Finally, 22 studies conducted from 2000 to 2023 were retained for data extraction based on prespecified inclusion and exclusion criteria (Figure 1).

The characteristics of the included studies are summarized in Table 1. The design of most studies was case-control [52,53,54,55,56,57,58,59,60,61,62], eight were cross-sectional [63,64,65,66,67,68,69,70], one was a cohort design [71], and two were randomised control trials [72,73]. Eighteen studies were conducted in the East African region: fourteen were conducted in Ethiopia [52,54,55,56,57,58,59,60,61,63,66,67,70,71] and one, per country, was conducted in Kenya [72], Rwanda [64], Uganda [73], and Somalia [62]. One was conducted in the North African region, Sudan [68]. One study was conducted in the West African region, Ghana [53]. One study was conducted in Zimbabwe, Southern Africa [65]. Additionally, one study involved multiple countries in the Sub-Saharan African region [69]. Two studies were conducted using the Demographic Health Survey (DHS) data, which is the most common health survey used in this region [64,69] (Table 1).

### 3.2. Participants

A total of *n*= 55,456 women who gave birth were included in this analysis. One study did not report the number of urban–rural participants [65], and of the remainder that did, 35,242 of 52,506 women (approximately 67%) resided in rural areas. Two studies were from malaria-endemic areas [72,73]. In 17 studies, selected participants attended healthcare facilities [52,53,54,55,56,57,58,59,60,61,62,63,66,67,68,70,73] (Table 1).

### 3.3. Interventions/Exposures

Eleven studies assessed the effects of IFA supplementation [52,55,56,57,59,62,64,68,70,71,73], whereas nine studies assessed the effects of iron-only supplementation [53,54,58,60,65,66,67,69,72], and three studies assessed the effect of folate-only supplementation [61,63,65]. One study analysed the separate effects of iron-only and folate-only supplementation [65]. Two included RCTs used a placebo for comparison [72,73], while the remaining studies did not include a relevant comparison group (Table 2).

Five studies assessed the duration and timing of supplementation [53,62,65,66,69]. For example, Adam et al. studied the effect of regularity of iron-only supplementation on low birth weight; however, they did not clearly define “always” and “irregular” iron supplementation [53]. Similarly, a study performed by Kumlachew et al. assessed the effects of taking iron-only supplementation for less than one month or two or more months on low birth weight [66]. Another study conducted by Traore et al. investigated the duration of iron-only supplementation, less than 60 days, 60 to 89 days, and 90 or more days, on low birth weight [69]. Additionally, a study by Omar et al. assessed the effects of IFA supplementation in each trimester on preterm birth [62] (Table 2).

Both of the included RCTs provided specific information about the dosage of the supplementation [72,73]. Ndyomugyen et al. administered 120 mg elemental iron and 5 mg folic acid supplementation as an intervention [73], while Mwangi et al. utilised 60 mg elemental iron supplementation [72]; however, neither of these trials assessed the dose–response effect (Appendix A). Even though the rest of the studies did not report the doses of iron and folic acid supplementation, it is important to note that the available and recommended doses are 60 mg elemental iron and 0.4 mg folic acid in many African countries, such as Ethiopia [74] and Sudan [75] (Table 2). 

### 3.4. Quality Assessment

The quality assessment we conducted showed that among all the included studies, nineteen were at low risk of bias [52,53,54,55,56,57,58,59,60,61,62,63,64,66,67,70,71,72,73] and three studies were at medium risk of bias [65,68,69] (Appendix A). 

All cross-sectional studies included information about the representativeness of the sample and its adequacy. Three studies obtained lower scores in the outcome assessment, primarily attributed to self-reporting, which might have impacted the internal validity of the studies [65,68,69]. Only two studies controlled for all the important variables in the final model (such as antenatal care, residence, maternal age, and educational status) [63,67]. In all case-control studies, details such as the definition and representativeness of cases, definition and selection of controls, and assessment of both cases and controls were provided. All studies noted reliance on maternal recall for collecting information on iron and folate supplementation status, potentially introducing recall bias. All studies are unmatched case-control studies (Appendix A). 

Both RCTs included adequate information about baseline and outcome measurements, random sequence generation, allocation of concealment, reporting bias and bias related to measurement, and validity [72,73] (Appendix A).

### 3.5. Outcomes (n = 22)

Among all the included studies, nineteen studies reported low birth weight [52,53,54,56,57,58,59,60,63,64,65,66,67,68,69,70,71,72,73] and four studies reported preterm birth [55,61,62,72]. One study conducted by Mwangi et al. [72] reported on both outcomes (Table 2).

#### 3.5.1. Low Birth Weight (*n* = 19)

Out of 19 studies, 15 studies reported that iron-only, folate-only, or IFA supplementation reduced the likelihood of low birth weight [52,53,54,56,57,59,60,63,64,67,68,69,70,71,72] (Table 2). 

However, three studies reported no statistically significant association between iron-only or IFA supplementation and low birth weight, suggesting that these supplements may not be effective in reducing the odds of low birth weight. In a clinical trial by Ndyomugyen et al. involving 860 rural primigravida mothers, no significant association was found between IFA supplementation and low birth weight (aRR 1.23; 95% CI: 0.85–1.78) [73], with 34% of participants in the intervention group. Similarly, a facility-based study by Mulu et al. that included 279 participants, of whom 80.6% took iron-only supplements, showed that iron-only supplementation did not significantly reduce the odds of low birth weight (aOR 0.60; 95% CI: 0.13 to 1.15) [58]. In a large-scale study by Traore et al. [69] involving 36,879 infant–mother pairs, where 90% received antenatal care and 80% took iron supplements, no significant association was found between iron-only supplementation and low birth weight, although the exact ORs and CIs were not reported (Table 2). 

Two studies [63,65] assessed the relationship between folate-only supplementation and low birth weight, with differing results. A study by Mehare et al., involving 472 full-term newborn–mother pairs, found that 52% of participants received antenatal care and 73% took folate-only supplementation. Their findings indicate that folate-only supplementation during pregnancy was associated with a significant reduction in the odds of low birth weight (aOR 5.48; 95% CI: 2.93 to 10.25) [63]. In contrast, a nationwide study conducted by Foto et al. in Zimbabwe, involving mother–newborn pairs, reported no significant association between folate-only supplementation and the reduction in low birth weight (cOR 0.95; 95% CI: 0.71 to 1.27) [65] (Table 2); however, it is important to note that the study did not report the proportions of participants who received antenatal services or took folate-only supplementation, which may limit the interpretation of these findings. 

Four examined the relationship between supplementation duration and regularity on low birth weight and showed mixed results. In a high-quality study by Adam et al. involving 360 mothers, irregular intake (*n* = 80) or non-intake (*n* = 36) of iron-only supplementation during pregnancy increased the odds of low birth weight compared to regular intake, although the results were not statistically significant (aOR 2.19; 95% CI: 0.99 to 4.84 and aOR 3.19; 95% CI: 0.99 to 4.84, respectively) [53]. Similarly, Kumlachew et al., in a study of 381 mother–neonate pairs, found that not taking iron-only supplementation (*n* = 53) increased the odds of low birth weight compared to taking it for more than two months (aOR 4.00; 95% CI: 1.30 to 12.60), while taking it for up to one month did not significantly affect the odds (*n* = 63; aOR 2.43; 95% CI: 0.83 to 7.17) [66]. Traore et al., in a community-based study of 36,879 infant–mother pairs, found no statistically significant difference in low birth weight reduction [69] (Table 2).

#### 3.5.2. Preterm Birth (*n* = 4)

Two high-quality studies found that IFA supplementation was associated with reduced odds of preterm birth [55,62]. In a facility-based study by Deriba involving 483 mothers, 23.6% did not take IFA supplementation during pregnancy. The findings indicate that not taking IFA supplements increased the odds of preterm birth (aOR 2.26; 95% CI: 1.22 to 4.18) [55] (Table 2). Similarly, Omar et al.’s study of 499 women reported that 34.5% who did not take IFA supplements had higher odds of a preterm birth (cOR 2.80; 95% CI: 1.67 to 4.68) [62]. Further analysis in Omar et al.’s study revealed that IFA supplementation during the first trimester (*n* = 46; cOR 0.51; 95% CI: 0.32 to 0.82), second trimester (*n* = 122; cOR 0.35; 95% CI: 0.18 to 0.69), and third trimester (*n* = 160; cOR 0.25; 95% CI: 0.12 to 0.52) was associated with decreased odds of preterm birth [62] (Table 2). 

In a clinical trial conducted by Mwangi et al. among 430 rural pregnant women, 215(50%) were randomised into the iron-only group. The findings indicated that iron-only supplementation reduced the risk of preterm birth by 7% (absolute risk difference (ARD) −7.00; 95% CI: −13.20 to −1.10) [72]. Another study by Wudie et al. included 288 women–newborn pairs, finding that 81% took folate-only supplementation and this reduced the odds of preterm birth by 74% [61] (Table 2). 

### 3.6. Meta-Analysis 

#### 3.6.1. IFA Supplementation on Low Birth Weight (Meta-Analysis) (*n* = 9)

All nine studies reporting IFA supplementation were included in this meta-analysis [52,56,57,59,64,68,70,71,73]. A total of 11,236 participants were included. We employed a random effect model to estimate the pooled effects of IFA supplementation on low birth weight. The odds of low birth weight among mothers who took IFA supplementation was 0.37 times that of non-users, indicating a 63% reduction (OR 0.37; 95% CI: 0.29 to 0.48). There is high heterogeneity found among the included studies (*I*^2^ = 89.5%) (Figure 2). Therefore, the findings should be interpreted with caution. The publication bias is assessed using both the funnel plot and Egger’s test. Egger’s test result showed no evidence of publication bias (*p*-value = 0.973). 

##### Subgroup Analysis of Iron Folate Supplementation and Low Birth Weight

Besides employing random effects in the meta-analysis, we conducted a subgroup analysis to explore potential heterogeneity based on the study design and country. 

Among nine studies included in the meta-analysis, three were cross-sectional [64,68,70], four were case-control [52,56,57,59], one was a cohort [71], and another one was a clinical trial by design [73]. We categorised all cross-sectional studies into the “cross-sectional” category, case-control studies into the “case-control” category, while cohort and clinical trials studies were grouped into “others”. The subgroup meta-analysis result showed that IFA supplementation reduced the odds of low birth weight by 66% among the cross-sectional group (OR 0.34; 95% CI: 0.24 to 0.50), and by 72% (OR 0.28; 95% CI: 0.24 to 0.34) in the case-control group. The effect of IFA supplementation on low birth weight in the “others” group was lower than in the cross-sectional and case-control groups but did not reach statistical significance, set at 0.05 (OR 0.62; 95% CI: 0.37 to 1.02) (Galbraith plot attached as Appendix A). The test of group difference showed that there was evidence of a difference in effect size between groups (*p*-value = 0.01). Moreover, consistently high heterogeneity was found among the cross-sectional (*I*^2^ = 79.61%) and the “others” (*I*^2^ = 88.73%) groups. However, no evidence of heterogeneity was found among the case-control group (*I*^2^ = 29.84%) (Figure 3).

We also conducted a subgroup analysis based on the study country. Among the nine included studies, six were conducted in Ethiopia [52,56,57,59,70,71] and the remaining three were conducted in Sudan, Uganda, and Rwanda [64,68,73]. Consequently, we grouped all studies conducted in Ethiopia under the “Ethiopia” category and those from Sudan, Uganda, and Rwanda grouped under the “others” category. The subgroup analysis based on the study country showed that IFA supplementation had a low effect in Sudan, Ruanda, and Uganda of 0.49 (OR 0.49; 95% CI: 0.28 to 0.87) compared to Ethiopia, 0.32 (OR 0.32; 95% CI: 0.25 to 0.41) (see Figure 4). However, the test of group differences showed that there was no difference between groups (*p* value = 0.17). High heterogeneity was identified in Ethiopia (*I*^2^ = 82%) and the “others” category (Sudan, Uganda, and Rwanda) (*I*^2^ = 91.45%) (Figure 4). Therefore, findings should be interpreted with caution. 

Furthermore, the subgroup analysis based on study setting also showed that IFA supplementation had a higher significant effect on the facility-based study of 0.34 (OR 0.34; 95% CI: 0.25 to 0.46) compared to community-based study settings, 0.49 (OR 0.49; 95% CI: 0.43 to 0.55). The test of group difference showed that there was a significant difference between groups (*p*-value = 0.03). Additionally, there was no evidence of heterogeneity identified in the community-based study (*I*^2^ = 0.00%), whereas high heterogeneity was identified in the facility-based study (*I*^2^ = 88.03%) (Figure 5). 

##### Sensitivity Test

A sensitivity test was conducted to explore individual study’s effects on the overall effect size for IFA. The results of the sensitivity test revealed that one study had a comparably large significant impact on the overall effect sizes compared to the other studies. The omission of the study conducted by Ndyomungyenyi et al. [73] from the analysis resulted in an increase in the overall effect size of IFA supplementation on low birth weight (OR 0.34; 95% CI: 0.27 to 0.41), indicating a reduction in the odds ratio by 0.035 (Table 3). 

#### 3.6.2. Effects of Iron-Only Supplementation on Low Birth Weight (Meta-Analysis) (*n* = 8)

A total of nine studies [53,54,58,60,65,66,67,69,72] reported the relationship between iron-only supplementation and low birth weight. Here, the study conducted by Traore et al. [69] was excluded from the meta-analysis due to not reporting the exact effect size. Consequently, this meta-analysis incorporated data from eight studies [53,54,58,60,65,66,67,72], comprising a total of 5605 participants. The results indicated that iron-only supplementation during pregnancy reduced the odds of low birth by 68% (OR 0.32; 95% CI: 0.21 to 0.50). Nevertheless, there was high heterogeneity found among the included studies (*I*^2^ = 99.16%) (Figure 6). Additionally, Egger’s test for publication bias showed evidence of publication bias among the included studies, with a *p*-value of 0.038. Therefore, the finding should be interpreted with caution, as heterogeneity and publication bias can impact the validity and generalizability of the finding.

##### Subgroup Analysis of Iron-Only Supplementation and Low Birth Weight

To explore the heterogeneity, we conducted a subgroup analysis based on study design and country. Among the eight studies included in this meta-analysis, four were case-control studies [53,54,58,60], three were cross-sectional studies [65,66,67], and one was a clinical trial [72]. We categorised all cross-sectional studies into the “cross-sectional” category, while the remaining case-control and clinical trial studies were grouped into the “others” category.

The subgroup analysis result showed that iron-only supplementation during pregnancy in the cross-sectional category had a higher effect on reducing low birth weight (OR 0.24; 95% CI: 0.08 to 0.76) compared to the “others” category (OR 0.36; 95% CI: 0.32 to 0.41). The test of group difference showed that the difference between groups was not statistically significant, with a *p*-value of 0.49. In the subgroup analysis, high heterogeneity was found in the cross-sectional group (*I*^2^ = 99.88%). In contrast, no evidence of heterogeneity was found among other groups (*I*^2^ = 0.0%), suggesting greater consistency in the findings within this subgroup (Figure 7). Differences in the heterogeneity emphasise the importance of considering the study design in interpreting the overall result.

Similarly, subgroup analysis was conducted based on countries where the studies were carried out. Studies conducted in Ethiopia were grouped under “Ethiopia” [54,58,60,66,67], while those conducted in Ghana, Zimbabwe, and Kenya were grouped under the “others” category [53,65,72]. The findings showed that the effect of iron-only supplementation on reducing low birth weight was high in Ethiopia (OR 0.27; 95% CI: 0.14 to 0.49) compared to the “others” category (Ghana, Zimbabwe, and Kenya) (OR 0.46; 95% CI: 0.29 to 0.73). However, the test of group difference was not statistically significant (*p*-value = 0.16). High heterogeneity was found in both groups (*I*^2^ = 99.55%) in Ethiopia and (*I*^2^ = 83.83%) in the “others” category (Figure 8). This suggests high variability in the study results within both country categories. This variability may be attributed to the differences in the characteristics of study participants between the groups. Among studies conducted in the Ethiopian groups, around 63.2% of the participants resided in urban areas, whereas in the other groups, 69.5% of the participants resided in rural areas. Additionally, 79.8% of the participants received antenatal care services and 71.9% took iron supplementation in the Ethiopian groups, while 95% of participants received antenatal care services in the other groups. However, the study conducted by Foto et al. [65] did not report any descriptive statistics and none of the studies reported the dose of iron supplementation. 

Additionally, the subgroup analysis based on the study setting shows that iron-only supplementation has a high effect in facility-based studies (OR 0.27; 95% CI: 0.16 to 0.45) compared to community-based study settings (OR 0.55; 95% CI: 0.33 to 0.89). There was evidence of high heterogeneity identified in both facility-based groups (*I*^2^ = 82.70%) and community-based groups (*I*^2^ = 99.32%) (Figure 9).

##### Sensitivity Test

A sensitivity test analysis was performed among the studies included in the meta-analysis for iron-only supplementation. The findings revealed that two studies highly affected the overall effect size of iron-only supplementation on low birth weight. Specifically, the omitted study conducted by Kumlachew et al. [66] resulted in a 0.07 decrease in the overall effect sizes (OR 0.39; 95% CI: 0.29 to 0.52), with a *p*-value of 0.00 (Figure 10); the study was characterised by a highly reported effect size and a comparably small sample size (381 participants). However, the study conducted by Foto et al. [65] led to a 0.03 increase in the overall effect sizes (OR 0.29; 95% CI: 0.18 to 0.45), with a *p*-value of 0.00 (Figure 10). Notably, the study performed by Foto et al. [65] is distinguished by a large sample size (2290 participants) and a crude effect size (they did not adjust confounders). Despite the removal of these two studies, the heterogeneity among the included studies remained high (*I*^2^ = 85.16%).

## 4. Discussion

This systematic review and accompanying meta-analysis aimed to comprehensively summarise and analyse the available evidence on the effects of iron and/or folate supplementation on low birth weight and preterm birth in Africa. Overall, our findings suggest that iron supplementation and IFA supplementation during pregnancy reduce the risk of low-birth-weight infants in this region. Similarly, IFA supplementation was associated with a reduced likelihood of preterm birth. 

The meta-analysis findings indicate that iron supplementation during pregnancy reduces the risk of low birth weight in African women, aligning with a 2013 global meta-analysis [16]—albeit with limited African representation. In addition, the meta-analysis conducted by Haider et al. [16] considered both oral iron supplementation and fortifications as an intervention. However, our updated review included the most recent publications and provides a region-specific analysis, addressing distinct factors influencing health outcomes in African pregnant women. The observed benefit may be attributed to iron’s direct involvement in cell proliferation and DNA synthesis, crucial processes for normal growth and development, especially during the intrauterine period [76]. 

This meta-analysis, consistent with previous systematic reviews [40], identified a reduction in low birth weight with IFA supplementation. Even though the findings aligned, the previous review was based on Western and Asian countries, including only one African country, whereas this review was based on more than nine studies in different African countries. However, the finding of this meta-analysis is different from another review conducted in 2012 [41], which found no statistically significant association between IFA and reductions in low birth weight. The reason for this difference might be due to the earlier global review being based on a single study and country conducted in England [77].

Iron is crucial for oxygen-carrying proteins like haemoglobin, myoglobin, and hem enzymes [78,79], ensuring an adequate oxygen supply for fetal development and normal growth [79]. Global systematic reviews and meta-analyses [80,81] demonstrate that iron alone and IFA supplementation reduce the anaemia burden, improving haemoglobin and ferritin levels. For communities facing high burdens of anaemia and food insecurity, studies suggest iron supplementation is a cost-effective intervention in reducing low birth weight [82,83], indicating it is important in mitigating the risk of low birth weight in the low-resource African context. Overall, the results of this meta-analysis indicate that both iron-only supplementation and IFA supplementation have comparable effects on reducing low birth weight within the African region. However, our data were insufficient to make conclusive findings.

The subgroup analysis for iron-only supplementation showed that the effects of iron-only supplementation during pregnancy were higher in Ethiopia, cross-sectional, and facility-based studies. The observed variation in the odds ratio might be due to differences in the baseline characteristics of the study participants, arising from geographical disparities, the prevalence of anaemia, social class, ethnic groups, and the socio-economic status of the respective countries. Nearly 73% of the participants in the Ethiopia category resided in urban areas, whereas a substantial 76.85% of the participants in the “others” category were from rural settings. Even among women who resided in rural areas, benefits were observed, albeit with high heterogeneity. Nevertheless, the findings are promising and should be considered by public health policymakers or health service providers in regions/countries where there are limited resources when designing programs and policies. Similarly, studies in Africa reported varying burdens of anaemia across different geographical locations, with rates such as 31.66% in Ethiopia [84], 42% in Ghana [33], and 57% in Kenya [85]. With this high rate of anaemia, only one study from Kenya that met the inclusion criteria was included in this review and the findings indicated a positive influence of iron-only supplementation on reducing low birth weight. Therefore, further research is needed in Kenya to understand the effects of iron-only supplementation on low birth weight. Additionally, Rono, Kombe, and Makokha (2018) conducted a study in Kenya that assessed the effects of MMN supplementation versus IFA supplementation on pica practices and haemoglobin levels. The study found that MMN supplementation during pregnancy was significantly associated with a reduction in pica practices but did not show a significant effect on haemoglobin levels compared to IFA [86]. This highlights the need for more research to explore the contextual factors influencing the effectiveness of different supplementation strategies in Kenya. Overall, the subgroup analysis highlights the importance of considering contextual factors when interpreting the impact of interventions across diverse populations. Moreover, women attending healthcare facilities had better health outcomes. Therefore, ensuring access to healthcare facilities is a crucial policy decision for women, especially those residing in rural areas. 

Similarly, the subgroup analysis showed that the effect size of IFA supplementation is higher in the “others” category (Sudan, Uganda, and Ruanda) and facility-based studies. The reason for this difference might be due to the time variation in the study period and the sample size difference among the categories. Studies in the “others” category were conducted before 2015, while studies in the Ethiopia category were conducted after 2015, this time variation may impose the difference in proportions of women who took the minimum recommended IFA tablets. For example, a study based on the 2014/15 Ruanda national survey reported that only 5.5% of pregnant women took the minimum recommended IFA supplementation [87], while in Ethiopia, 41.38% of pregnant women took the minimum recommended IFA supplementation in 2020 [88]. Additionally, differences in sample size among the included studies could be another possible reason, for instance, the mean sample size of studies conducted in the “others” category was 3032, while the mean sample size of studies conducted in the Ethiopia category was 357. A large sample size provides more reliable estimates by increasing precision and statistical power [89], which may influence the observed effect size. Moreover, high heterogeneity was observed among the facility-based studies, while community-based studies showed consistent findings. This variation may be attributed to differences in the methodological approach, institutional context, and population characteristics such as demographics in facility-based studies. Facility-based studies often involve more diverse populations with varying health conditions and access to care, which can introduce variability in outcomes. Although reliance on self-reports can introduce biases such as recall bias or social desirability bias in both settings, these biases may have greater impacts in facility-based studies due to the more heterogeneous populations and varying levels of interaction with health care providers. In contrast, community-based studies typically involve populations with more uniform demographic characteristics, contributing to their lack of variability. 

Two studies, one conducted in Zimbabwe and the other in Ethiopia [63,65], presented inconsistent findings regarding the association between folate-only supplementation and low birth weight. The disparity in results could be attributed to variations in the characteristics of the study participants, differences in sample size, and the extent of confounder adjustment; for example, the study conducted by Mehare et al. [63] involved predominantly urban residents, comprising nearly 80% of the participants, whereas the majority (70%) of the participants in the study by Foto et al. [65] resided in rural areas. Additionally, the study conducted by Foto et al. [65] reported a crude association but none of the socio-demographic, clinical, and health service-related variables were controlled, while these variables were controlled in the study conducted by Mehare et al. [63]. This discrepancy in methodology could have contributed to the different results between the studies. Finally, there was a substantial sample size difference between the studies, with 472 participants in Mehare et al. [63] and 3221 in Foto et al. [65]. 

Furthermore, four studies [53,65,66,69] differently reported the association between the duration and regularity of iron-only and IFA supplementation on low birth weight. The reason for this variation may be attributed to the differences in how the exposure variable was categorized in each study. For instance, the study by Adam et al. categorized the exposure as “not at all”, “irregularly”, and “always” [53]. On the other hand, Kumlachew et al. categorized exposure as “not at all”, “less than one month”, and “greater than two months” [66]. Following a different approach, Traore et al. categorized the exposure into “not at all”, “less than 60 tablets”, “60 to 90 tablets”, and “greater than 90 tablets” [69]. It is noteworthy that the World Health Organisation (WHO) recommends pregnant women receive IFA supplementation throughout their pregnancy [26], whereas the majority of African countries implement a minimum of 3 months (or 90 tablets) of iron-only or IFA supplementation during pregnancy [90,91]. Unfortunately, the included studies in this review utilised different categorizations to measure the effects of the duration of the supplementation and low birth weight and we could not develop one consistent exposure measurement using the information provided in each article. These findings highlight the importance of standardizing categories of exposure variables to better understand the dose/frequency that may contribute to improvements in low birth weight for women residing in the African region. 

This review identified two studies [55,62] that consistently reported a positive association between IFA supplementation and reduced preterm birth. Additionally, iron-only, and folate-only supplementation are also associated with the odds of reducing preterm birth [61,72]. Despite the available evidence not permitting the estimation of the effect size of iron-only, folate-only, or IFA supplementation on preterm birth, this finding suggests that such supplementation during pregnancy does reduce the odds of preterm birth in the African region. The effect sizes of IFA and folate-only supplementation are aligned but our evidence is insufficient to compare the effectiveness of the IFA and folate-only supplementation. Therefore, IFA, iron-only, and folate-only supplementation all have a positive effect on reducing the risk of preterm birth further. 

In regard to the appropriate methodological implications, such as confounder adjustments, inconsistencies were observed across the studies included in this review. Among all the studies included in this review, twelve adjusted for anaemia and/or pregnancy complications as a confounder in the final model, even though anaemia and pregnancy complications had an intermediate effect on the causal pathway. The presence of a mediator variable can change the observed effect sizes [92] of the relationship between iron-only, folate-only, or IFA supplementation on adverse birth outcomes, making it difficult to draw clear conclusions. Additionally, four studies did not make any adjustments for confounders or other important variables in the model, and four studies omitted adjustments for socio-demographic variables. Furthermore, none of the studies conducted stratification analysis by age, ethnicity, antenatal care utilisation, education, or rurality. 

The geographical scope of the included studies is mostly focused on Ethiopia. This may be attributed to the rapid growth of higher educational and public health institutes, from the federal to zonal level, leading to an increase in research outputs within the country.

The exclusion criteria aim to maintain a specific focus on the impact of oral iron and folate supplementation during pregnancy by excluding studies involving fortification, combinations with other micronutrients, and those measuring serum iron or folate levels as interventions. This exclusionary approach is adopted to improve comparability and minimize confounding factors, thereby ensuring a clearer understanding of the isolated effects of oral iron-only, folate-only, or IFA supplementation. Also, Africa is a resource-poor region; therefore, we are interested in interventions that are accessible in such settings.

This systematic review and meta-analysis have been subjected to different limitations. The first is the high heterogeneity among the included studies, which is expected given the diversity across the African continent. This heterogeneity introduces uncertainty and makes it necessary to interpret the results cautiously, as they may not be universally applicable across different settings. Additionally, including studies with cross-sectional designs limits the ability to establish clear causal relationships between iron-only or IFA supplementation and low birth weight. A significant limitation is the reliance on maternal self-reporting in many of the included studies. This method increases the risk of socially desirable responses, recall bias, and potential misreporting of supplement intake, which could lead to either over- or underestimation of actual consumption. Such inaccuracies may result in an incomplete or misleading understanding of the true relationship between supplementation and birth outcomes. Furthermore, restricting the included studies to those published in English may have excluded relevant research published in other languages, potentially introducing publication bias. Acknowledging this limitation is essential for a nuanced interpretation of the findings and highlights areas for improvement and refinement in future research. 

## 5. Conclusions

Iron-only and IFA supplementations are effective in reducing low birth weight in Africa. In addition, IFA supplementation has a positive impact on reducing preterm birth. However, the existing evidence is insufficient to establish a clear relationship between folate-only supplementation and low birth weight. Considering these findings, we recommend conducting further primary studies that specifically address the limitations identified among rural women with limited support and low levels of literacy by aiming to provide health policymakers with fundamental data necessary for the implementation of interventions and strategies. Additionally, future studies should consider the geographical disparity and the quality of healthcare services (including the availability and readiness of nutritional services). Overall, the findings highlight the importance of the continued promotion and implementation of these interventions in maternal health programs in Africa. Therefore, key strategies for promoting the uptake of iron and IFA supplementation are strongly recommended, including ongoing professional development to frontline health workers in ANC services who can facilitate behaviour change to improve women’s knowledge of and access to supplementation, which evidence shows can protect women and their babies from the preventable, adverse birth outcomes.

## Figures and Tables

**Figure 1 nutrients-16-02801-f001:**
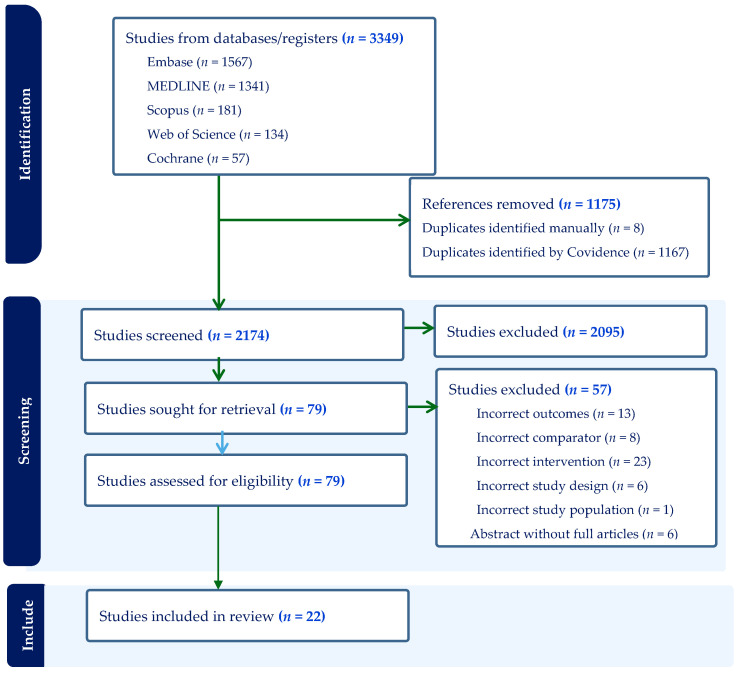
PRISMA (Preferred Reporting Items for Systematic Review and Meta-Analysis) flow chart diagram of studies searched for systematic review and meta-analysis in Africa, 2023.

**Figure 2 nutrients-16-02801-f002:**
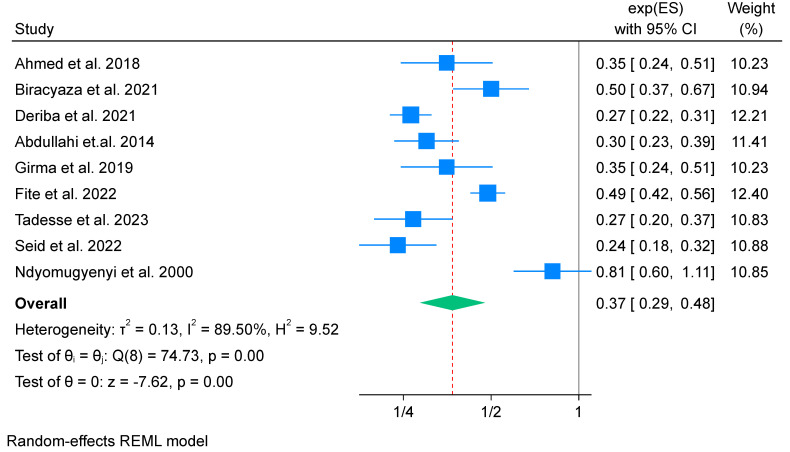
Forest plot of overall effects of IFA supplementation on low birth weight in Africa 2023 [52,56,57,59,64,68,70,71,73].

**Figure 3 nutrients-16-02801-f003:**
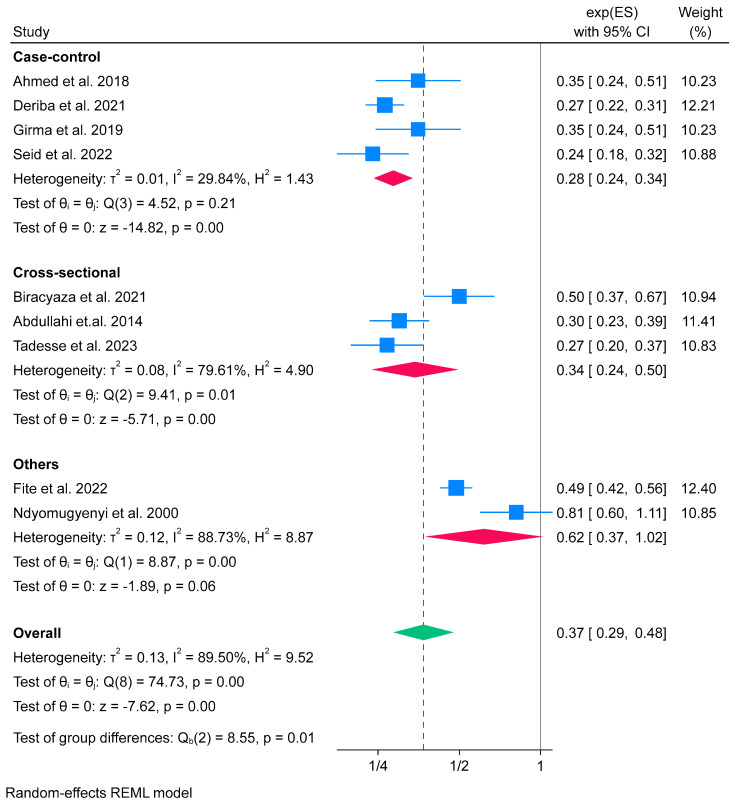
Subgroup analysis of IFA supplementation on low birth weight by study design in Africa, 2023 [52,56,57,59,64,68,70,71,73].

**Figure 4 nutrients-16-02801-f004:**
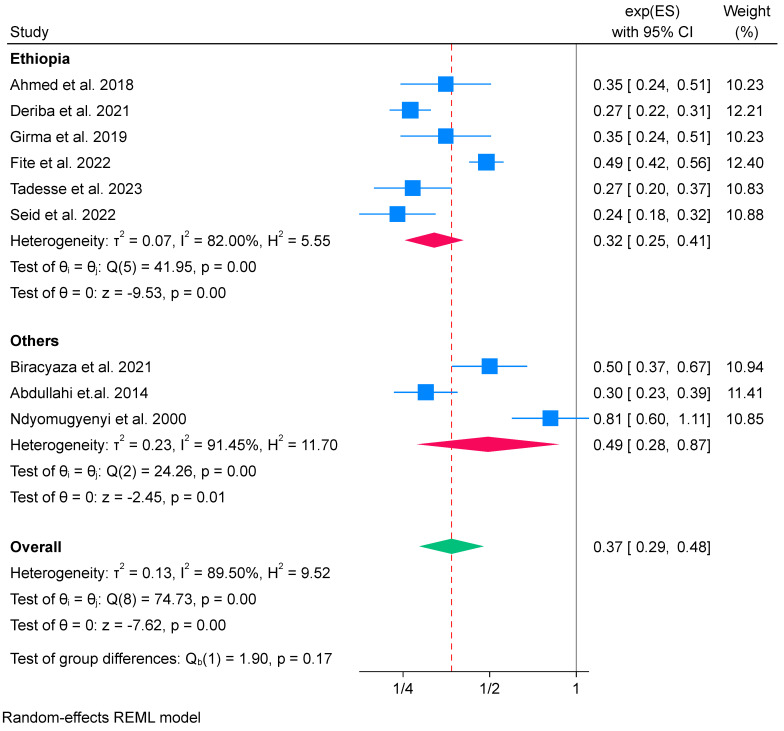
Subgroup analysis of IFA supplementation on low birth weight by study country in Africa, 2023 [52,56,57,59,64,68,70,71,73].

**Figure 5 nutrients-16-02801-f005:**
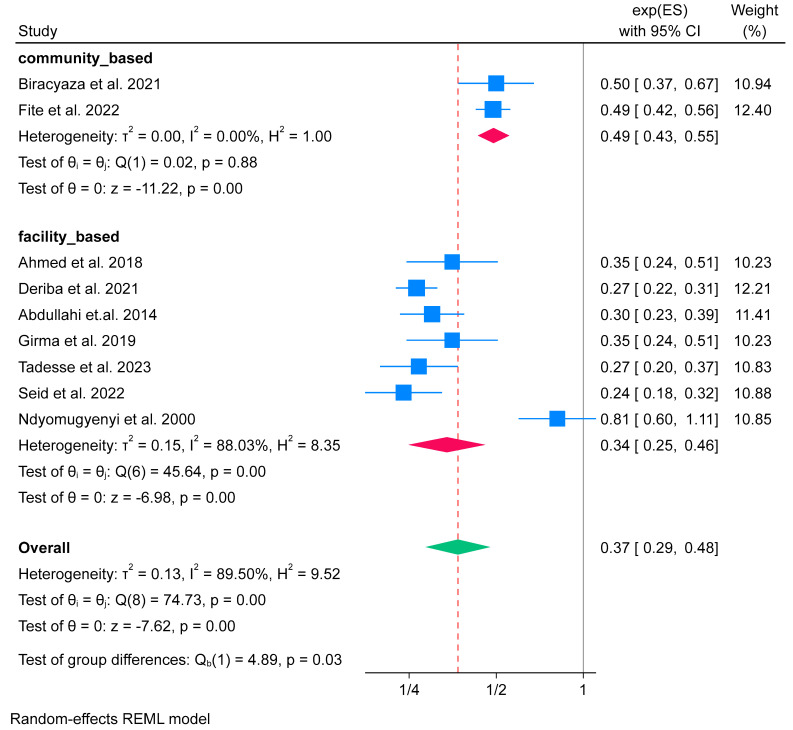
Subgroup analysis of IFA supplementation on low birth weight by study settings in Africa, 2023 [52,56,57,59,64,68,70,71,73].

**Figure 6 nutrients-16-02801-f006:**
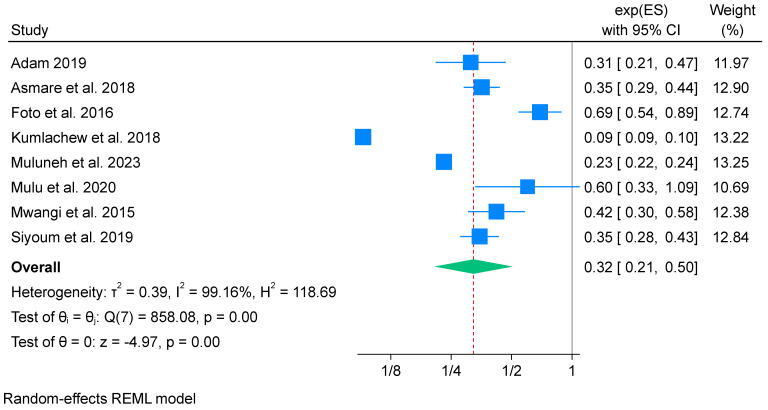
Forest plot of overall effects of iron-only supplementation on low birth weight in Africa 2023 [53,54,58,60,65,66,67,72].

**Figure 7 nutrients-16-02801-f007:**
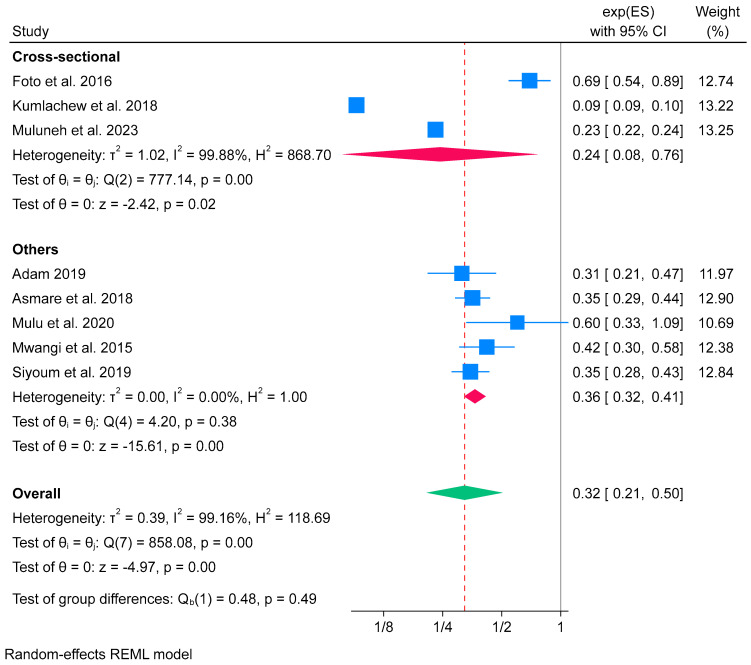
Subgroup analysis of iron-only supplementation on low birth weight by study design in Africa, 2023 [53,54,58,60,65,66,67,72].

**Figure 8 nutrients-16-02801-f008:**
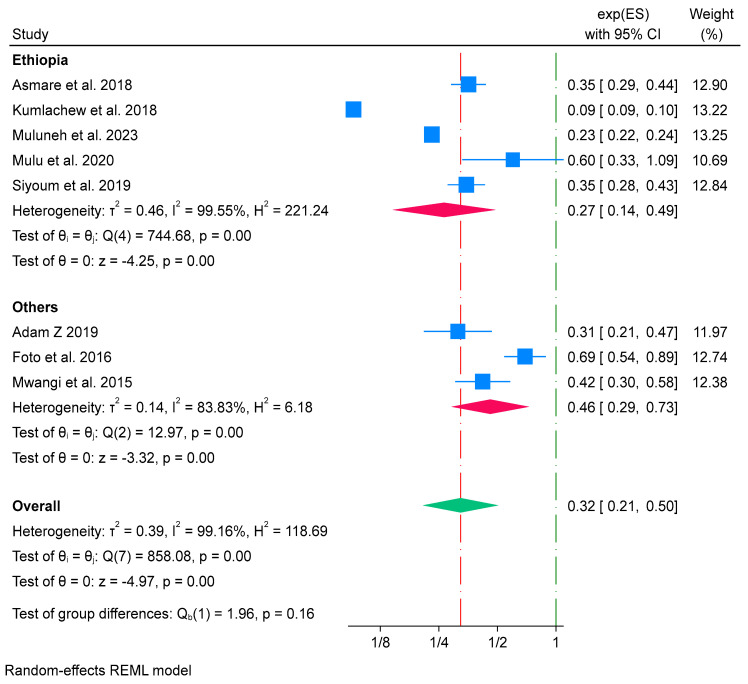
Subgroup analysis of iron-only supplementation on low birth weight by study country in Africa, 2023 [53,54,58,60,65,66,67,72].

**Figure 9 nutrients-16-02801-f009:**
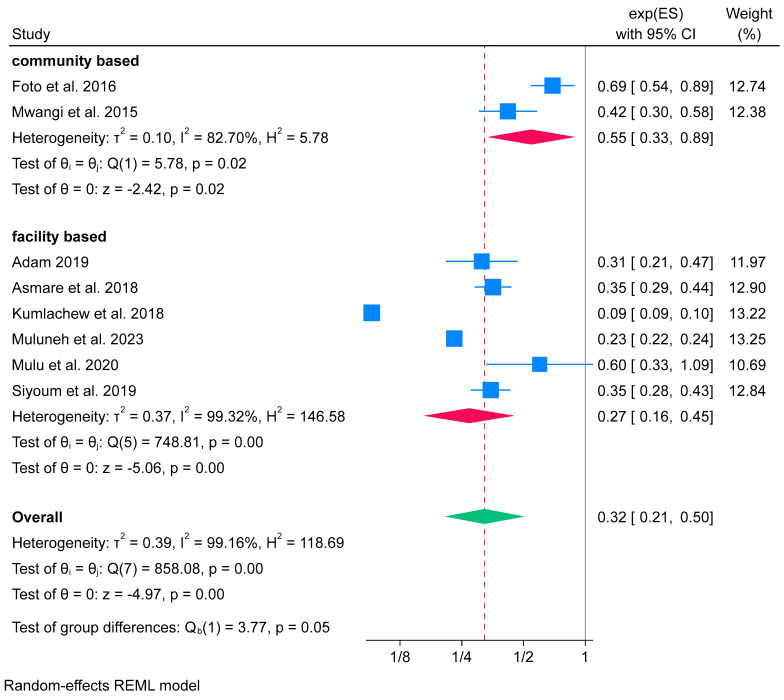
Subgroup analysis of iron-only supplementation on low birth weight by study setting in Africa, 2023 [53,54,58,60,65,66,67,72].

**Figure 10 nutrients-16-02801-f010:**
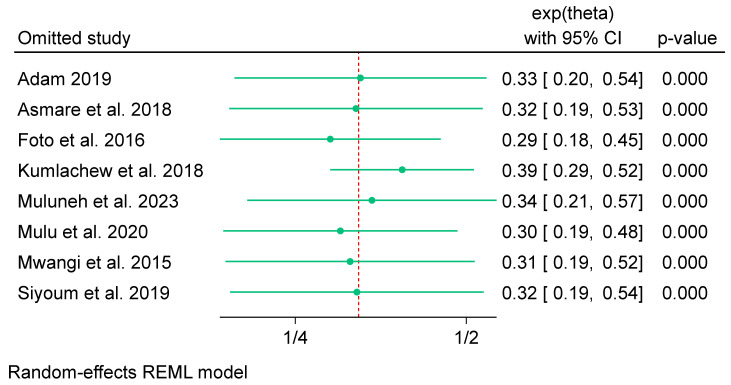
Sensitivity test results of iron-only supplementation on low birth weight among studies included for meta-analysis, Africa 2023 [53,54,58,60,65,66,67,72].

**Table 1 nutrients-16-02801-t001:** Characteristics of included articles conducted in the African region from 2000 to 2023.

#	First Author, Pub. Year (Ref)	Country (Setting)	Design/(Case to Control Ratio)	Study Population	Year Study	Sample Size (% Rural)	Exposure Assessment	Outcome Assessment
1.	Ahmed 2018 [52]	Ethiopia (Facility-based)	Unmatched Case-control (1 to 2)	Mother–newborn pairs.	2017	286 (40.9%)	Self-reported	Birth weight was assessed within one hour using a digital Seca scale, and then categorized as low birth weight (less than 2.5 kg) or normal birth weight (2.5 kg or greater).
2.	Biracyaza 2021 [64]	Rwanda (Community-based)	Cross-sectional	Mother–newborn pairs.	2014/2015	7381 (82.3%)	Self-reported	Birth weight was measured using metric units (in kilograms) and subsequently classified as low birth weight (less than 2.5 kg) or normal birth weight (2.5 kg or greater).
3.	Deriba 2021 [56]	Ethiopia (Facility-based)	Unmatched Case-control (1 to 2)	Mother–newborn pairs.	2020	555 (56%)	Self-reported	Birth weight was assessed within one hour using a digital Seca scale and categorized as either low birth weight (less than 2.5 kg) or normal birth weight (2.5 kg or greater).
4.	Abdullahi 2014 [68]	Sudan (Facility-based)	Cross-sectional	Mother–newborn pairs.	2012	856 (38.7%)	Self-reported	Birth weight was extracted from the client’s card and categorized as either low birth weight (less than 2.5 kg) or normal birth weight (2.5 kg or greater).
5.	Girma 2019 [57]	Ethiopia (Facility-based)	Unmatched Case-control (1 to 2)	Mother–newborn pairs.	2017	279 (41.6%)	Self-reported	Birth weight was determined using a balanced Seca scale with precision to the nearest 0.01 g and categorized as low birth weight (less than 2.5 kg) or normal birth weight (2.5 kg or greater).
6.	Fite 2022 [71]	Ethiopia (Community-based)	Prospective Cohort	Mother–newborn pairs.	2021	427 (100%)	Self-reported	Birth weight was assessed using a calibrated Docbel BRAUNH scale, rounded to the nearest 100 g, and subsequently categorized as either low birth weight (less than 2500 g) or normal (2500 g or greater).
7.	Tadesse 2023 [70]	Ethiopia study (Facility-based)	Cross-sectional	Mother–newborn pairs	2021	337 (8.6%)	Self-reported	A birth weight below 2500 g was classified as low birth weight.
8.	Seid 2022 [59]	Ethiopia (Facility-based)	Unmatched Case-control (1 to 2)	Mother–newborn pairs	2020	255 (70.2%)	Self-reported	Birth weight was recorded within one hour using a Salter weight measurement scale and then categorized as low birth weight (less than 2.5 kg) or normal birth weight (2.5 kg or greater).
9.	Ndyomugyenyi 2000 [73]	Uganda (Facility-based)	Double-blind, randomized trial	Mother–newborn pairs	1996 to 1998	860 (100%)	By researchers	For babies born in health facilities, birth weight was measured immediately after delivery, while for those born in the community, only birth weights measured within 7 days after delivery were included. Birth weight of less than 2.5 kg was classified as low birth weight, and normal birth weight was defined as greater than or equal to 2.5 kg.
10.	Deriba 2021 [55]	Ethiopia (Facility-based)	Unmatched Case-control (1 to 2)	Mother–newborn pairs	2020	483 (54.55%)	Self-reported	Preterm birth was determined by examining the last menstrual period from an ANC card or an early period ultrasound. It was then categorized as preterm (birth occurring before 37 completed weeks) or normal (birth occurring after 37 completed weeks and within 42 weeks of gestation).
11.	Omar 2022 [62]	Somalia (Facility-based)	Case-control (approximately 1 to 6)	Mother–newborn pairs	2021	499 (3.4%)	Self-reported	Preterm birth, defined as birth occurring before 37 weeks of gestation, was assessed by a physician or midwife.
12.	Adam 2019 [53]	Ghana (Facility-based)	Case-control (1 to 2)	Mother–newborn pairs	2015 to 2016	360 (33.1%)	Self-reported	Birth weight was assessed within 24 h after delivery using standard weight measurement techniques and categorized as low birth weight (birth weight less than 2500 g) or normal birth weight (birth weight between 2500 g and 3400 g).
13.	Asmare 2018 [54]	Ethiopia (Facility-based)	Unmatched Case-control (1 to 2)	Mother–newborn pairs	2017	453 (40.8%)	Self-reported	The outcome was determined within 15 min using a standard scale. A newborn weighing less than 2500 g was classified as having a low birth weight, while a weight greater than 2500 g was considered a normal birth weight.
14.	Foto 2016 [65]	Zimbabwe (Community-based)	Cross-sectional	Mother–newborn pairs	2014	2950 (not reported)	Self-reported	Birth weight was evaluated by reviewing the client card or self-report, and categorized as low birth weight if it was less than 2500 g; otherwise, it was considered normal.
15.	Kumlachew 2018 [66]	Ethiopia (Facility-based)	Cross-sectional	Mother–newborn pairs	2018	375 (33.3%)	Self-reported	Birth weight was measured after 30 min using a balanced weight scale and categorized as low birth weight if it was less than 2.5 kg; otherwise, it was considered normal.
16.	Muluneh 2023 [67]	Ethiopia (Facility-based)	Cross-sectional	Mother–newborn pairs	2021	422 (63.27%)	Self-reported	A standard weight scale was utilised, and a newborn weighing less than 2500 g was classified as having a low birth weight, while those weighing between 2500 g and 4000 g were considered normal.
17.	Mulu 2020 [58]	Ethiopia (Facility-based)	Unmatched Case-control (1 to 2)	Mother–newborn pairs	2019	279 (all from urban area)	Self-reported	Birth weight was measured within 1–2 h after birth using a balanced Seca Scale and categorized as low birth weight if it was less than 2.5 kg; otherwise, it was considered normal.
18.	Siyoum 2019 [60]	Ethiopia (Facility-based)	Unmatched Case-control (1 to 2)	Mother–newborn pairs	2018	330 (33.9%)	Self-reported	Birth weight was assessed using a calibrated Seca scale and rounded to the nearest 100 g. It was then categorized into low birth weight (less than 2500 g) and normal (between 2500 g and 4000 g).
19.	Mwangi 2015 [72]	Kenya (Community-based)	Double-blind randomized controlled trial	Mother–newborn pairs	2011 to 2013	430 (all from rural area)	By researcher	Birth weight was measured immediately after delivery or upon presentation at a research clinic within 7 days for those who delivered at home. Women whose newborns had a birth weight of less than 2500 g were classified as having a low birth weight.
								Preterm birth, defined as birth occurring before 37 weeks of gestation, was determined by a physician or midwife.
20.	Traore 2023 [69]	Sub-Saharan Africa (Community-based)	Cross-sectional	Mother–infant pairs	2014 to 2020	36,879 (68%)	Self-reported	Birth weight was evaluated through a record review (card) and classified as low birth weight (birth weight less than 2500 g) or normal (birth weight greater than or equal to 2500 g).
21.	Mehare 2020 [63]	Ethiopia (Facility-based)	Cross-sectional	Mother–newborn pairs	2018 to 2019	472 ((21.8%)	Self-reported	Birth weight was measured within 24 h using a standard weight scale with precision to the nearest 1 g and categorized as low birth weight (less than 2500 g) or normal birth weight (2500 g or greater).
22.	Wudie 2019 [61]	Ethiopia (Facility-based)	Unmatched Case-control (1 to 2)	Mother–newborn pairs	2016	288 (34%)	Self-reported	Preterm birth, assessed by gestational age (birth of the newborn before 28 weeks of gestation), was documented by referring to the client’s records.

Footnote: g: gram, kg: kilogram.

**Table 2 nutrients-16-02801-t002:** Findings of included studies conducted in the African region from 2000 to 2023.

Low Birth Weight							
#	Author (Year)	Exposure	Outcome	Statistical Method	Effect Estimates (OR, RR, HR) (95% CI)		Confounder Adjustment
1.	Ahmed 2018 [52]	IFA supplementation	Low birth weight	Binary logistic regression	IFA	Ref	Maternal age, educational status of the mother, occupation of mother, height, MUAC, any multivitamin, anaemia, ANC follow-up, nutritional counselling, additional food intake during pregnancy, MDD-W, khat chewing, history of abortion, history of preterm delivery, HIV status, type of pregnancy, and infant’s sex.
					No	aOR 2.84 (1.15 to 7.03)	
2.	Biracyaza 2021 [64]	IFA supplementation	Low birth weight	Binary logistic regression	No	Ref	Residence, sex of newborn, maternal education, maternal age, TT vaccine, BMI, gestational age, smoking during pregnancy, ANC utilisation, family size, household wealth index, anaemia, marital status, counselling about nutrition, provided information about complications, and head of the household.
					IFA	aOR 0.50 (0.30 to 0.90)	
3.	Deriba 2021 [56]	IFA supplementation	Low birth weight	Binary logistic regression	IFA	Ref	Educational status, nutrition counselling, additional meals, restriction of foods during pregnancy, MUAC of the mother, height of the mother, frequency of eating, ANC for recent pregnancy, anaemia status of the mother, medical illness on recent pregnancy, pregnancy-related complications, drinking alcohol, and gestational age in weeks.
					No	aOR 3.78 (2.10 to 6.85)	
4.	Abdullahi 2014 [68]	IFA supplementation	Low birth weight	Binary logistic regression	No	Ref	None of the variables were controlled.
					IFA	cOR 0.30 (0.17 to 0.68)	
5.	Girma 2019 [57]	IFA supplementation	Low birth weight	Binary logistic regression	IFA	Ref	Nutritional counselling, taking snacks during pregnancy, maternal MUAC, anaemia, and minimum dietary diversity score of women.
					No	aOR 2.84 (1.15 to 7.03)	
6.	Fite 2022 [71]	IFA supplementation	Low birth weight	Poisson regression	No	Ref	Occupation of the women, sex of the neonate, maternal nutritional status, fertility desire, maternal height, and time to health facilities.
					IFA	aPR 0.55 (0.36 to 0.84)	
7.	Tadesse 2023 [70]	IFA supplementation	Low birth weight	Binary logistic regression	No	Ref	Respondents’ age, history of stillbirth, previous history of LBW, history of chronic medical illness, haemoglobin level, history of ANC, follow-up, sex of newborn, and extra meals during their pregnancy.
					IFA	aOR 0.27 (0.10 to 0.72)	
8.	Seid 2022 [59]	IFA supplementation	Low birth weight	Binary logistic regression	IFA	Ref	Additional food, haemoglobin, food security, and MDD-W score.
					No	aOR 4.17 (1.44 to 12.30)	
9.	Ndyomugyenyi 2000 [73]	IFA supplementation	Low birth weight	Linear regression	IFA	Ref	Timing of ANC.
					No	aRR 1.23 (0.85 to 1.78)	
10.	Adam 2019 [53]	Iron-only supplementation	Low birth weight	Binary logistic regression	Iron-only	Ref	Planned pregnancy, mode of delivery, parity, and previous low birth weight.
					No	aOR 3.20 (1.10 to 9.50)	
					Always	Ref	
					Irregular intake	aOR 2.19 (0.99 to 4.63)	
					No intake	aOR 3.19 (0.99 to 4.63)	
11.	Asmare 2018 [54]	Iron-only supplementation	Low birth weight	Binary logistic regression	Iron-only	Ref	Sex of the newborn, residence, educational status, MUAC category, history of abortion, ANC visit, complications during pregnancy, parity, gestational age, and history of LBW.
					No	aOR 2.82 (1.62 to 4.91)	
12.	Foto 2016 [65]	Iron-only supplementation	Low birth weight	Bivariable analysis	No	Ref	None of the variables were controlled.
					Iron-only	cOR 0.69 (0.49 to 0.98)	
			Low birth weight	Bivariable analysis	Duration of iron-only supplementation	cOR 0.89 (0.82 to 0.97)	
		Folate-only supplementation	Low birth weight	Bivariable analysis	No	Ref	
					Folate	cOR 0.95 (0.71 to 1.27)	
13.	Kumlachew 2018 [66]	Iron-only supplementation	Low birth weight	Binary logistic regression	Iron more than 2 months	Ref	Marital status, residence, educational status, occupation, ethnicity, parity, maternal MUAC, history of abortion, number of ANC, malaria, PIH, anaemia, previous LBW, confirmed DM, maternal weight, maternal height, substance use, maternal abuse, gestational age (weeks), and sex of neonate.
					Up to 1 month	aOR 2.43 (0.83, 7.17)	
					Not taken	aOR 4.00 (1.30 to12.60)	
14.	Muluneh 2023 [67]	Iron-only supplementation	Low birth weight	Binary logistic regression	No	Ref	Mother’s age, residence, marital status education, occupation, alcohol consumption during pregnancy, smoking during pregnancy, history of spontaneous abortion, ANC visit, DM during pregnancy, history of hypertension, and history of anaemia.
					Iron-only	aOR 0.23 (0.20 to 0.25)	
15.	Mulu 2020 [58]	Iron-only supplementation	Low birth weight	Binary logistic regression	No	Ref	Mother’s education, parity, pregnancy complications, gestational hypertension, nutritional counselling, height, age, anaemia, additional meal intake, BMI, incomplete ANC visit, maternal MUAC, pre-pregnancy weight, and gravidity.
					Iron-only	aOR 0.60 (0.30 to 1.50)	
16.	Siyoum 2019 [60]	Iron-only supplementation	Low birth weight	Binary logistic regression	Iron-only	Ref	Age of the mother, MUAC, GA, occupation, presence of complications during pregnancy, nutritional counselling, residence of the mother, level of education, ethnicity of the mother, age at first birth, age of the mother, and diseases.
					No	aOR 2.89 (1.58 to 5.29)	
17.	Mwangi 2015 [72]	Iron supplementation	Low birth weight	Multiple logistic regression	No	Ref	Gravidity, maternal age, HIV infection, plasmodium infection status, haemoglobin concentration, and gestational age.
					Iron-only	aOR 0.42 (0.13 to 0.78)	
18.	Traore 2023 [69]	Iron-only supplementation	Low birth weight	Binary logistic regression	No	Ref.	None of the variables were controlled.
					Iron < 60 days	NS (results not reported)	
					Iron 60 to 89 days	NS (Results not reported)	
					Iron 90 or more days	NS (Results not reported)	
19.	Mehare 2020 [63]	Folate-only supplementation	Low birth weight	Binary logistic regression	Folate-only	Ref	Maternal age, residency, educational status, occupation, marital status, birth interval, pregnancy type, ANC follow-up, dietary counsel, alcohol drinking, and cigarette smoking.
					No	aOR 5.48 (2.93 to 10.25)	
Preterm birth							
20.	Deriba 2021 [55]	IFA supplementation	Preterm birth	Binary logistic regression	IFA	Ref	Family size, education of mother, education of husband, occupation of mother, occupation of husband, nutritional counselling, taking additional meals, restriction of food, frequency of taking DGLV, meal frequency, and MUAC of mother.
					No	aOR 2.26 (1.22 to 4.18)	
21.	Omar 2022 [62]	IFA supplementation	Preterm birth	Bivariable logistic regression	IFA	Ref	None of the variables were controlled.
					No	cOR 2.80 (1.67 to 4.68)	
					No	Ref	
					1st trimester	cOR 0.51 (0.32 to 0.82)	
					2nd trimester	cOR 0.35 (0.18 to 0.69)	
					3rd trimester	cOR 0.25 (0.12 to 0.52)	
22.	Mwangi 2015 [72]	Iron-only supplementation	Preterm birth	Binary logistic regression	No	Ref	Gravidity, maternal age, HIV infection, plasmodium infection status, haemoglobin concentration, iron deficiency, and gestational age.
					Iron-only	ARD −7.00 (−13.20 to −1.10)	
23.	Wudie 2019 [61]	Folate-only supplementation	Premature delivery	Binary logistic regression	No	Ref	Residence, monthly income, occupation, level of education, gynaecological problems, number of antenatal clinic visits, age at first delivery (years), number of pregnancies, history of preterm birth, history of multiple pregnancies, history of stillbirth, nutritional counselling during pregnancy, additional nutrition during pregnancy, activity during pregnancy, and support during pregnancy.
					Folate-only	aOR 0.26 (0.008 to 0.084)	

Footnote: NB: adverse birth outcomes (preterm birth or stillbirths or perinatal death or neonatal death or abortion or low birth weight). ANC: antenatal care, MUAC: middle upper arm, IFA: iron folic acid, MDD-W: minimum dietary diversity-women, DM: diabetes mellitus, PIH: pregnancy-induced hypertension, LBW: low birth weight, GA: gestational age, BMI: body mass index, aOR: adjusted odds ratio, aRR: adjusted relative risk, cOR: crude odds ratio, aHR: adjusted hazard ratio, ARD: absolute risk difference.

**Table 3 nutrients-16-02801-t003:** Sensitivity test results of iron folate supplementation on low birth weight among studies included for meta-analysis, Africa 2023.

Omitted Study	Exp (Theta)	95% CI	*p* Value
Ahmed et al., 2018 [52]	0.371	0.278, 0.494	0.000
Biracyaza et al., 2021 [64]	0.355	0.269, 0.469	0.000
Deriba et al., 2021 [56]	0.386	0.293,0.509	0.000
Abdullahi et al., 2014 [68]	0.378	0.284, 0.509	0.000
Girma et al., 2019 [57]	0.371	0.278, 0.494	0.000
Fite et al., 2022 [71]	0.354	0.267, 0.470	0.000
Taddess et al., 2023 [70]	0.383	0.290, 0.505	0.000
Seid et al., 2022 [59]	0.388	0.298, 0.507	0.000
Ndyomugyenyi et al., 2000 [73]	0.335	0.274, 0.410	0.000
Exp (theta)	0.37	0.29, 0.48	0.000

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
