# Peer review of "Is Oral Iron and Folate Supplementation during Pregnancy Protective against Low Birth Weight and Preterm Birth in Africa? A Systematic Review and Meta-Analysis"

_nutrients, 2024, doi:10.3390/nu16162801_

Round 1
Reviewer 1 Report
Comments and Suggestions for Authors
Dear Authors,
You have done a laborious job of collecting and analyzing materials reflecting the use iron-only, folate-only or Iron-Folic Acid (IFA) oral supplementation on pregnancy outcomes in African countries. Your professional approach to analyzing the material deserves sincere respect. At the same time, please consider the following recommendations, which will help improve the article.
1. Please, in the “Introduction” section consider the possibility of citing and discussing the results presented in the following publication: Peña-Rosas JP, De-Regil LM, Dowswell T, Viteri FE. Daily oral iron supplementation during pregnancy. Cochrane Database Syst Rev. 2012 Dec 12;12:CD004736. doi: 10.1002/14651858.CD004736.pub4. Update in: Cochrane Database Syst Rev. 2015 Jul 22;(7):CD004736. doi: 10.1002/14651858.CD004736.pub5. PMID: 23235616; PMCID: PMC4233117), which analyzes the effects of daily oral use of iron supplements by pregnant women, either alone or in conjunction with folic acid or with other vitamins and minerals as a public health intervention.
Also, please, consider possibility of citing and discussing of the following study: Keats EC, Oh C, Chau T, Khalifa DS, Imdad A, Bhutta ZA. Effects of vitamin and mineral supplementation during pregnancy on maternal, birth, child health and development outcomes in low- and middle-income countries: A systematic review. Campbell Syst Rev. 2021 Jun 26;17(2):e1127. doi: 10.1002/cl2.1127. PMID: 37051178; PMCID: PMC8356361. This publication give the possibility to discuss the recommended iron doses for malaria-endemic areas and to compare effects of individual supplements and combinations of supplements on pregnancy outcomes. This discussion is quite appropriate since your study included studies from malaria-endemic areas [74, 75].
2. A citation of the article: «Betsy Chebet Rono, Yeri Kombe, Anselimo Makokha. (2018). Multiple Micronutrients Versus Iron Folic Acid on Pica and Hemoglobin Levels Among Pregnant Women in Kenya. Central African Journal of Public Health, 4(4), 95-101. https://doi.org/10.11648/j.cajph.20180404.11» could be included in the Discussion section and discuss the specifics of this study in relation to yours.
3. Please adjust the numbering of your sections so that there is no confusion. For example, section 1. ‘Materials and Methods’, followed by section 2.1. «Selection Criteria»; 3.1. Participants (246); 3.1. Interventions/exposures (252); 3.1. Quality assessment (276); 3.1. Outcomes (n=22) (293); 3.1.1. Low birth weight (n=19) (297); 3.1.1. Preterm birth (n=4) (348); 3.1.1. Meta-analysis (369); 3.1.1.1. IFA supplementation on low birth weight (Meta-analysis) (n=9) (370)4 3.1.1.1. Subgroup analysis of iron folate supplementation and low birth weight (382); 3.1.1.1. Sensitivity test (423); 3.1.1. Effects of iron-only supplementation on low birth weight (Meta-analysis) (n=8) (433); 3.1.1.1. Sensitivity test (492); 3. Discussion (507).
4. Please, in Figure 1. place the inscriptions in the blocks within their boundaries
5. Please, check the contents of the columns in Table 1. From the content of the column ‘Study population’, rows 1,7,8,14,16,17, follows that Neonates, Nnewborn, Children born during the two years before the survey - are the source of the Self-reported (column -Exposure assessment).
6. Please, discuss the weaknesses of self-reports and possible reasons why there was no evidence of heterogeneity identified in the community-based study, whereas high heterogeneity was identified in the facility-based study
7. Section 3.1. Outcomes (n=22) is difficult to read. Please do not repeat the data of the Table 2 in the text. The reader needs a clear interpretation of these data.
Best of luck in improving the manuscript!

Author Response
Response to Reviewer Comments
|
Thank you very much for taking the time to review this manuscript. Please find a detailed responses below and the revisions/corrections highlighted/in track changes in the re-submitted files.
Reviewer 1: |
Comments 1: Please, in the “Introduction” section consider the possibility of citing and discussing the results presented in the following publication: Peña-Rosas JP, De-Regil LM, Dowswell T, Viteri FE. Daily oral iron supplementation during pregnancy. Cochrane Database Syst Rev. 2012 Dec 12;12:CD004736. doi: 10.1002/14651858.CD004736.pub4. Update in: Cochrane Database Syst Rev. 2015 Jul 22;(7):CD004736. doi: 10.1002/14651858.CD004736.pub5. PMID: 23235616; PMCID: PMC4233117), which analyzes the effects of daily oral use of iron supplements by pregnant women, either alone or in conjunction with folic acid or with other vitamins and minerals as a public health intervention. Also, please, consider possibility of citing and discussing of the following study: Keats EC, Oh C, Chau T, Khalifa DS, Imdad A, Bhutta ZA. Effects of vitamin and mineral supplementation during pregnancy on maternal, birth, child health and development outcomes in low- and middle-income countries: A systematic review. Campbell Syst Rev. 2021 Jun 26;17(2):e1127. doi: 10.1002/cl2.1127. PMID: 37051178; PMCID: PMC8356361. This publication give the possibility to discuss the recommended iron doses for malaria-endemic areas and to compare effects of individual supplements and combinations of supplements on pregnancy outcomes. This discussion is quite appropriate since your study included studies from malaria-endemic areas [74, 75]. |
Response 1: Thank you for strengthening our manuscript. Based on your suggestion, we have discussed and cited the recommended articles in the revised version. “[Over a decade later, other reviews (40, 41) reported that iron supplementation reduced the risk of low birth weight but had no effect on preterm birth. However, a 2012 review indicated that IFA supplementation was not associated with a reduction in either low birth weight or preterm births (41). A systematic review conducted in 2013 (15) included oral iron-only supplementation and fortification as exposures. The review's findings showed that iron-only supplementation reduced low birth weight with no measured effect on preterm birth. A more recent systematic review conducted in 2021 in low- and middle-income countries found that iron supplementation, compared to placebo, and IFA supplementation, compared to folic acid alone, reduced the risk of low birth weight. Additionally, this review reported that multiple micronutrients (MMN) supplementation was more effective than iron with or without folate in reducing the risk of low birth weight (44). Past reviews showed mixed findings and did not comprehensively include countries in Africa. For instance, a systematic review conducted in 2012 (40) included only one study from Gambia, and a 2013 review (15) similarly included just one study from Zimbabwe. This implies there is a need to further explore the effects of iron-only, folate-only, and IFA supplementation on preterm birth and low birth weight, particularly in Africa.]” Line numbers 97 to 114
|
Comments 2: A citation of the article: «Betsy Chebet Rono, Yeri Kombe, Anselimo Makokha. (2018). Multiple Micronutrients Versus Iron Folic Acid on Pica and Hemoglobin Levels Among Pregnant Women in Kenya. Central African Journal of Public Health, 4(4), 95-101. https://doi.org/10.11648/j.cajph.20180404.11» could be included in the Discussion section and discuss the specifics of this study in relation to yours.
Response 2: Thank you for your insightful suggestion. We have corrected and cited the recommended article in the revised manuscript. “[Additionally, Rono, Kombe, and Makokha (2018) conducted a study in Kenya that assessed the effects of MMN supplementation versus IFA supplementation on pica practices and haemoglobin levels. The study found that MMN supplementation during pregnancy was significantly associated with a reduction in pica practices(eating nonfood items) but did not show a significant effect on haemoglobin levels compared to IFA (91). This highlights the need for more research to explore the contextual factors influencing the effectiveness of different supplementation strategies in Kenya.]” line numbers 565 to 571
Comments 3: Please adjust the numbering of your sections so that there is no confusion. For example, section 1. ‘Materials and Methods’, followed by section 2.1. «Selection Criteria»; 3.1. Participants (246); 3.1. Interventions/exposures (252); 3.1. Quality assessment (276); 3.1. Outcomes (n=22) (293); 3.1.1. Low birth weight (n=19) (297); 3.1.1. Preterm birth (n=4) (348); 3.1.1. Meta-analysis (369); 3.1.1.1. IFA supplementation on low birth weight (Meta-analysis) (n=9) (370)4 3.1.1.1. Subgroup analysis of iron folate supplementation and low birth weight (382); 3.1.1.1. Sensitivity test (423); 3.1.1. Effects of iron-only supplementation on low birth weight (Meta-analysis) (n=8) (433); 3.1.1.1. Sensitivity test (492); 3. Discussion (507).
Response 3: Thank you for your thorough review. We have revised and corrected all the numbers throughout the manuscript
Comments 4: Please, in Figure 1. place the inscriptions in the blocks within their boundaries
Response 4: Thank you for your suggestion. Following your comment, we grouped the blocks in the revised manuscript. Line numbers 232 to 250.
Comments 5: Please, check the contents of the columns in Table 1. From the content of the column ‘Study population’, rows 1,7,8,14,16,17, follows that Neonates, Nnewborn, Children born during the two years before the survey - are the source of the Self-reported (column -Exposure assessment).
Response 5: Thank you for your valuable input. We have corrected the study population and exposure assessment columns as suggested. Line numbers 269 to 270
Comments 6: Please, ddiscuss the weaknesses of self-reports and possible reasons why there was no evidence of heterogeneity identified in the community-based study, whereas high heterogeneity was identified in the facility-based study
Response 6: Thank you. We have included the weaknesses and possible reasons in the revised manuscript.
“[A significant limitation is the reliance on maternal self-reporting in many of the included studies. This method increases the risk of socially desirable responses, recall bias, and potential misreporting of supplement intake, which could lead to either over- or underestimation of actual consumption. Such inaccuracies may result in an incomplete or misleading understanding of the true relationship between supplementation and birth outcomes.]” line numbers 670 to 675
“[Moreover, high heterogeneity was observed among the facility-based studies, while community-based studies showed consistent findings. This variation may be attributed to differences in methodological approach, institutional context and population characteristics such as demographics in facility-based studies. Facility-based studies often involve more diverse populations with varying health conditions and access to care, which can introduce variability in outcomes. Although reliance on self-reports can introduce biases such as recall bias or social desirability bias in both settings, these biases may have greater impacts in facility-based studies due to the more heterogeneous populations and varying levels of interaction with health care providers. In contrast, community-based studies typically involve populations with more uniform demographic characteristics, contributing to their lack of variability.]” line numbers 590 to 600.
Comments 7: Section 3.1. Outcomes (n=22) is difficult to read. Please do not repeat the data of the Table 2 in the text. The reader needs a clear interpretation of these data. Response 7: Thank you for your suggestion. We have now substantially revised section between lines 319 to 371.
|

Reviewer 2 Report
Comments and Suggestions for Authors
The authors carried out a systematic review and meta-analysis on oral iron and folate supplementation during pregnancy. While several meta-analyses exist on the topic, the choice to focus on Africa and on relatively recent years clearly justifies this new approach. The major finding of the study is the protective effect of supplementation on low birthweight and, partly, on preterm birth.
Methods of EBM were meticulously applied during the study. The number of data sources included into the mathematical analysis appears to be sufficient to support the conclusions drawn.
The MS is convincingly written. The rules of publishing systematic reviews are carefully followed. The depth of the discussion correspond to the extent of analytical data provided.
Author Response
Thank you very much for taking the time to review this manuscript.
Reviewer 3 Report
Comments and Suggestions for Authors
Thank you for giving me the opportunity for review the manuscript entitled “Is Oral Iron and Folate Supplementation During Pregnancy Protective Against Low Birth Weight and Preterm Birth in Africa? A systematic Review and Meta-analysis.”
The manuscript is interesting and in scope of the Journal however it requires some clarifications.
The topic is of interest, as pregnant women have special dietary requirements that need to be met to prevent damage to the growing fetus as well as to prevent the development of certain diseases later in live.
Please find the specific comments below:
1. It is important that this systematic review and meta-analysis follows the Preferred Reporting Items for Systematic Review and Meta-Analysis (PRISMA-P) 2020 guidelines.
2. In the Discussion section of the manuscript, it should be more strongly emphasised that over- or underconsumption of iron-only, folate-only or IFA may occur when data are collected by questionnaire, which may not reflect the actual situation.
3. The methodological part of the manuscript should be improved by adding more information (description) on socio-demographic variables, education, age of mothers (if possible).
Author Response
Response to Reviewer Comments
|
Thank you very much for taking the time to review this manuscript. Please find a detailed responses below and the revisions/corrections highlighted/in track changes in the re-submitted files.
Reviewer 2: |
Comments 1: It is important that this systematic review and meta-analysis follows the Preferred Reporting Items for Systematic Review and Meta-Analysis (PRISMA-P) 2020 guidelines. Response 1: Thank you for your comments. The systematic review and meta-analysis followed the Preferred Reporting Items for Systematic Review and Meta-analysis (PRISMA-P) 2020 guidelines. The guideline with the corresponding page numbers is attached as a supplementary file.
Comments 2: In the Discussion section of the manuscript, it should be more strongly emphasised that over- or underconsumption of iron-only, folate-only or IFA may occur when data are collected by questionnaire, which may not reflect the actual situation.
Response 2: Thank you for your insightful suggestion. We mentioned the recommended information in the revised manuscript. “[A significant limitation is the reliance on maternal self-reporting in many of the included studies. This method increases the risk of socially desirable responses, recall bias, and potential misreporting of supplement intake, which could lead to either over- or underestimation of actual consumption. Such inaccuracies may result in an incomplete or misleading understanding of the true relationship between supplementation and birth outcomes.]” line numbers 670 to 675.
Comments 3: The methodological part of the manuscript should be improved by adding more information (description) on socio-demographic variables, education, age of mothers (if possible). Response 3: Thank you for your valuable comment. In the revised manuscript, we have included the age ranges of the participants and the proportions of rural residents. Unfortunately, the included studies assessed educational levels in different ways, which made it difficult for us to summarise that information. “[The participants were women aged 15 to 49 years, with approximately 67% of them residing in rural areas.]”, line numbers 142 to 143
|

Reviewer 4 Report
Comments and Suggestions for Authors
This outstanding article examines nutrient supplementation during pregnancy in Africa. The paper is structured and comprehensive, based on sufficient databases (despite the generally low number of studies). I have some comments:
Statistics: in the non-significant Forest plots, a Galbraith plot could be used as a support. It is indeed quite striking that vitamin B9 alone would not have any consequences on birth term nor birth weight.
Discussion: Africa is a large and diverse continent; could this explain the studies' heterogeneity? This could imply that your results are not universally applicable to the whole continent.
Author Response
Response to Reviewer Comments
|
Thank you very much for taking the time to review this manuscript. Please find a detailed responses below and the revisions/corrections highlighted/in track changes in the re-submitted files.
|
Reviewer three Comments 1: Statistics: in the non-significant Forest plots, a Galbraith plot could be used as a support. It is indeed quite striking that vitamin B9 alone would not have any consequences on birth term nor birth weight. Response 2: Thank you for your valuable comment. We have included the following Galbraith plot as a supplementary file in the revised manuscript, line numbers 398 to 399.
Figure S1. Galbraith plots for analysis of heterogeneity between studies in Africa. Comments 2: Discussion: Africa is a large and diverse continent; could this explain the studies' heterogeneity? This could imply that your results are not universally applicable to the whole continent. Response 2: Thank you for your suggestion. We have included the following information in the revised manuscript. “[This systematic review and meta-analysis are subjected to different limitations. The first is the high heterogeneity among the included studies, which is expected given the diversity across the African continent. This heterogeneity introduces uncertainty and, making it necessary to interpret the results cautiously, as they may not be universally applicable across different settings.]” line numbers 664 to 668.
|
